# Fanconi Anaemia, Childhood Cancer and the *BRCA* Genes

**DOI:** 10.3390/genes12101520

**Published:** 2021-09-27

**Authors:** Emma R. Woodward, Stefan Meyer

**Affiliations:** 1Manchester Centre for Genomic Medicine, Manchester University Hospitals NHS Foundation Trust, Manchester M13 9WL, UK; 2Division of Evolution and Genomic Sciences, School of Biological Sciences, Faculty of Biology, Medicine and Health, University of Manchester, Manchester M13 9WL, UK; 3Manchester Academic Health Science Centre, Manchester M13 9WL, UK; 4Department of Paediatric Haematology and Oncology, Royal Manchester Children’s Hospital, Manchester NHS Foundation Trust, Manchester M13 9JW, UK; 5Department of Paediatric and Adolescent Oncology, The Christie NHS Foundation Trust, Manchester M20 4BX, UK; 6Division of Cancer Studies, Faculty of Medical and Human Sciences, University of Manchester, Manchester M20 4GJ, UK

**Keywords:** fanconi anaemia, *BRCA1*, *BRCA2*, childhood cancer

## Abstract

Fanconi anaemia (FA) is an inherited chromosomal instability disorder characterised by congenital and developmental abnormalities and a strong cancer predisposition. In less than 5% of cases FA can be caused by bi-allelic pathogenic variants (PGVs) in *BRCA2/FANCD1* and in very rare cases by bi-allelic PGVs in *BRCA1/FANCS*. The rarity of FA-like presentation due to PGVs in *BRCA2* and even more due to PGVs in *BRCA1* supports a fundamental role of the encoded proteins for normal development and prevention of malignant transformation. While FA caused by *BRCA1/2* PGVs is strongly associated with distinct spectra of embryonal childhood cancers and AML with *BRCA2*-PGVs, and also early epithelial cancers with *BRCA1* PGVs, germline variants in the *BRCA1/2* genes have also been identified in non-FA childhood malignancies, and thereby implying the possibility of a role of *BRCA* PGVs also for non-syndromic cancer predisposition in children. We provide a concise review of aspects of the clinical and genetic features of *BRCA1*/*2*-associated FA with a focus on associated malignancies, and review novel aspects of the role of germline *BRCA2* and *BRCA1* PGVs occurring in non-FA childhood cancer and discuss aspects of clinical and biological implications.

## 1. Introduction

Fanconi anaemia (FA) is a complex inherited chromosomal instability disorder with congenital and developmental abnormalities, and cancer predisposition [1,2]. The classical clinical phenotype of FA presents with microcephaly, thumb and radial ray abnormalities, short stature, and café au lait spots. Progressive bone marrow failure is in the majority of cases observed in the first and second decade of life. However, affected individuals can also present with a more subtle phenotype and haematological and neoplastic complications in their third and fourth decades, or with a severe phenotype in need of complex interventions for multiple congenital malformations and malignant complications early in life [2,3,4,5]. After its initial description in 1927 [6], FA has been recognised as the most common paediatric bone marrow failure syndrome. FA-associated bone marrow failure can transform to acute myeloid leukaemia (AML), which is often encountered in later childhood, the second or even third decade. However, with better outcome of haematopoietic stem cell transplantation, solid tumours have become more prevalent in FA, in particular squamous cell carcinoma (SCC) affecting the aerodigestive and anogenital tract [1,5]. FA can be caused by germline pathogenic gene variants (PGVs) in at least 22 FA genes, *FANCA-FANCW*, of which all but the X-linked *FANCB* are autosomal, and all but *FANCR/RAD51* inherited recessively [5,7,8,9]. On a cellular level, the FA-pathway is firmly positioned in the context of DNA repair. The FA-genes encode proteins that form a network in the DNA damage response and have a major role in stabilising replication forks and mitigating replication stress, in particular from cross linker-induced, and aldehyde metabolism-generated genotoxic stress [8,10,11]. In this network multiple FA proteins (including FANCA, B, C, E, F, G, L) form the FA-core complex, which facilitates the ubiquitination of the FANCD2 and FANCI proteins [8,12]. Downstream of the FANCD2/FANCI ubiquitination, the other FA-proteins, including FANCD1/BRCA2 (herein referred to only as BRCA2) and FANCS/BRCA1 (herein referred to as BRCA1) have a more direct DNA-interaction with a role in homology-directed DNA repair [13]. The DNA damage response defect in FA is exploited diagnostically by demonstration of increased cross linker sensitivity in FA patient-derived cells [4]. The discovery of *BRCA2* as the gene of which bi-allelic PGVs can cause FA of the complementation group FA-D1 [14], and, more recently, PGVs in *BRCA1* for the FA complementation group FA-S [15] has placed FA from a rare paediatric genetic disease firmly in the context of familial cancer. In turn, recent large genetic studies of individuals affected by apparently sporadic childhood cancer detected *BRCA1/2* variants, which point also towards a possible role for germline PGVs in particular of *BRCA2* in the development of some non-syndromic childhood cancers. Here we review aspects of the clinical, genetic, and biological role of *BRCA1* and *BRCA2* in the context of FA and childhood cancer and discuss clinical and biological implications. 

## 2. FA Caused by *FANCD1*/*BRCA2* Pathogenic Variants: Severe Phenotype with Early Embryonal Malignancies

BRCA2 has an essential role in development, and homology-directed recombination and repair, and interstrand cross link response [13,16]. Whilst women with germline PGVs of *BRCA2* are at increased risk of hereditary breast and ovarian cancer (HBOC) [17], bi-allelic *BRCA2*-disruption was for a long time considered embryonically lethal. This was supported by evidence from mouse models, in which total absence of BRCA2 function appeared not compatible with foetal viability [18,19,20]. However, in 2002 *BRCA2* germline PGVs were identified in four children with FA of the complementation group FA-D1 [14]. Since then, more than 50 individuals with FA caused bi-allelic *BRCA2* PGVs have been reported in diverse ethnic groups [21,22,23,24,25,26,27,28], and *BRCA2* PGVs are considered to cause around 3–5% of FA [1]. Clinically, FA caused by bi-allelic *BRCA2* PGVs does not have an obvious sex preference and is in most cases associated with a severe phenotype, often with clinical features in the combination of the VACTER-L (vertebral, anal, oesophageal, cardiac renal, and radial dysplasia) complex [5,29]. Importantly, most of these patients develop aggressive malignancies and sometimes multiple cancers very early in life, with a cancer incidence over 90% at the age of 5 years [30]. In contrast to the more common defects in the FA-core complex genes including *FANCA*, *FANCC* or *FANCG*, the malignancies in *BRCA2*-associated FA include in addition to early AML also Wilms’ Tumour, embryonal brain tumours, (mostly medulloblastoma but also glioblastoma), hepatoblastoma, T-cell acute lymphoblastic leukaemia (ALL), neuroblastoma, and two cases of individuals with rhabdomyosarcoma have been reported [21,22,23,24,25,26,27,31,32,33]. One family had an index case with early onset colorectal carcinoma but an otherwise milder phenotype, and also lymphoma, breast cancer and AML was diagnosed in the family [28]. In a single individual of Turkish origin homozygosity for a hypomorphic missense *BRCA2* variant and primary ovarian insufficiency presented without clinical features of FA and an abrogated cellular phenotype [34]. SCCs have not been reported associated with BRCA2-PGV- associated FA. In FA the spectrum of pathogenic variants in *BRCA2* comprises splice-site variants, small deletions and insertions, and missense mutations. Several pathogenic variants were identified in more than one pedigree in the published cohort of FA-D1 cases. These include IVS7 splice site pathogenic variants, the 886delGT, and the Ashkenazi Jewish (AJ) founder mutation 6174delT. Additionally, the mutations 3492insT and 9424C>T have each been identified in multiple pedigrees. However, only very few *BRCA2* PGVs have been found homozygous in FA-D1 patients. These include IVS7 splice site PGVs, IVS19-1 G>A, 1548del4 and c.1538del4, c.1310del4 in exon 10, c.8524C>T and c.469A>T [21,24,26,34]. Importantly, the common AJ-founder mutation 6174delT [35], or the Icelandic founder mutation 999del5 [36] have not been observed in the homozygous state, implying that not all *BRCA2* PGVs provide for foetal viability, and that at least one allele needs to provide some essential BRCA2 function for survival. Despite the relatively small number of reported cases of FA caused by *BRCA2* PGVs, and multiple different cancers in individuals with the same mutations, there appears to be some association of individual PGVs with distinct cancer types in affected children; individuals with variants affecting the IVS7 splice site region appear to develop preferentially AML and not CNS malignancies, while none of the 6174delT and 886delGT carriers have been reported to develop AML, but rather, in most cases, brain tumours [21,33]. Detailed analysis of *BRCA2* PGVs associated with FA have provided insights into genetic and biological mechanisms of cellular and organismal survival conveyed by alleles containing *BRCA2* PGVs. This work suggests a major role for alternative spliced *BRCA2*-transcripts, which splice out variant-harbouring regions [28,37,38]. The alternatively spliced transcripts have been shown to encode variant spliced BRCA2 proteins, which have been demonstrated to maintain BRCA2 functional properties, some of which entirely proficient in DNA repair [37,38,39,40].

## 3. FA Caused by *BRCA1/FANCS* Pathogenic Variants: Distinct Clinical Phenotype and Cancer Spectrum

After the identification of *BRCA2* as the gene for which bi-allelic pathogenic variants underlie the mostly severe FA-phenotype in the complementation group FA-D1, in 2015 also pathogenic variants in the other main HBOC gene, *BRCA1*, were identified to cause an FA-like syndrome [15]. Individuals of this small but important subgroup of only 10 reported patients display in most cases cellular cross linker hypersensitivity and have clinical features of developmental abnormalities, and severe cancer predisposition [15,41]. *BRCA1* was therefore also termed *FANCS* [15]. Both BRCA1 and BRCA2 are essential in the homologous recombination repair pathway of double-stranded DNA (ds-DNA) breaks (reviewed in detail elsewhere [13,42]), with distinct but complementary roles, which for both proteins include a specific function for the protection against aldehyde toxicity [43]. As such BRCA1 is recruited early in the process to facilitate assembly of the complexes that signal the presence of DNA damage and promotes resection of DNA leading to the formation of single stranded DNA at the end of ds-DNA break. BRCA1 then recruits the PALB2 protein (Partner and Localisator of BRCA2) [44], which in turn recruits BRCA2 to facilitate assembly of RAD51 to provide a homologous template for DNA synthesis and repair [13,42,45]. *PALB2* is another gene of which bi-allelic germ line PGVs can cause FA, also termed *FANCN*, with a similar clinical presentation and cancer spectrum as FA caused by *BRCA2* PGVs including childhood solid tumours [46,47]. Total absence of BRCA1 protein function does also not appear to be compatible with foetal viability, as also evident by mouse studies [20,48,49]. This points to its essential role for normal early development, which is also reflected in the extreme rarity of the condition. Individuals carrying bi-allelic *BRCA1* PGVs must therefore have residual BRCA1 protein function, which at least in some cases is also provided by alternative splicing of the *BRCA1* gene [47]. To date there are 10 individuals of variable ethnic background with bi-allelic *BRCA1* PGVs reported [15,41,47,50,51,52,53], of which intriguingly only two are male [41,47]. The mutation spectrum of PGVs BRCA1 is homozygous in six cases from three pedigrees, all of which affect exon 11 [47,51]. The remaining individuals are compound heterozygous. While some, but not all, of the reported individuals affected by bi-allelic *BRCA1* PGVs have developmental abnormalities, such as microcephaly and intrauterine growth retardation (IUGR), which are also commonly found associated with classical FA, other FA-typical clinical findings, such as radial ray abnormalities, are not a frequent feature of the FA-like presentation of bi-allelic *BRCA1* PGVs (Figure 1). Developmental delay with mild to moderate learning difficulties, which is not typical for classical FA, is frequently described associated with bi-allelic *BRCA1* PGVs [41]. Importantly, bone marrow failure and transformation to AML, which is a classical clinical hallmark of FA, has not been reported with FA-like clinical features caused by bi-allelic *BRCA1* PGVs. While the cancer spectrum encountered by individuals affected by bi-allelic *BRCA1* PGVs includes also two individual cases of T-cell ALL and neuroblastoma [41,47], which are encountered in *BRCA2*-associated and also in *FANCN/PALPB2* -associated FA [46], the other reported individuals have developed more characteristic HBOC-associated adult-type epithelial malignancies early in life, such as breast and ovarian cancer [41,51]. The phenotype associated with BRCA1 deficiency does therefore appear distinct from the more classic form of FA and also FA associated with BRCA2-deficiency (Figure 1). In particular the absence of bone marrow failure and immunodeficiency might place BRCA1deficiency distinctly between FA and other chromosomal instability syndromes [2,41].

## 4. *BRCA1* and *BRCA2* PGVs in Non-FA Childhood Cancer 

Initial studies investigating a potential role of germline *BRCA1* and *BRCA2* PGVs also for predisposition for non-syndromic childhood cancer focussed on the incidence of childhood cancers in the offspring of families presenting with HBOC, in which a germline *BRCA1*/*2* PGV had been detected. When compared with non-*BRCA1*/*2* HBOC families no excess of childhood cancer was demonstrated in the families with a germline *BRCA1*/*2* PGV [54]. However, comparison with population controls detected an increased occurrence in childhood cancers in families with a *BRCA2* PGV, but not *BRCA1* [55]. Whilst not demonstrating causality, as the germline *BRCA2* status of affected children was not known and the possibility that there may be linked variation in modifier genes in affected families as demonstrated for HBOC *BRCA2* families [56], this was the first suggestion of an association of *BRCA2* PGVs with childhood cancer predisposition. With the advent of modern genomic technologies enabling large volume and high throughput sequencing, it became feasible to screen large childhood cancer populations for PGVs affecting multiple cancer predisposition genes. Two large scale studies identified germline *BRCA2* PGVs in a small number (13/2081 total across both studies) of childhood cancers and just one germline *BRCA1* PGV [57,58]. In the St Jude’s cohort of long-term survivors of childhood cancer germline *BRCA1/2* PGVs were detected in 34 (*BRCA2 n* = 20, *BRCA1 n* = 14)/4402 individuals [59]. Discovery studies of germ line sequence variants of individuals affected by specific tumour types that are more frequent in the childhood and younger adult population have also detected a small number of individuals mostly with germline *BRCA2* PGVs in osteosarcoma [60], paediatric glioma [61] and rhabdomyosarcoma [62]. However, given the population frequency of germline *BRCA1/2* PGVs, which is now being considered to be ~1 in 250 with an approximately 2:3 ratio of *BRCA1*/*2* [63], supportive evidence for a causative association in clinical and diagnostic settings not typical of the classical HBOC cancer spectrum is important, for example through case-control studies. For this, a relative risk of at least 4 was demonstrated for germline *BRCA*2 PGVs in a case-control study of medulloblastoma [64], an odds ratio (OR) of 3.6 for rhabdomyosarcoma [65], and an OR of 5.0 for paediatric and adolescent non-Hodgkin lymphoma, which intriguingly is not part of the cancer spectrum in classical HBOC and BRCA2-associated FA [66] (Table 1). Thus, despite *BRCA1* and *BRCA2* PGVs being not uncommon in the general population and therefore likely to be detected in large scale discovery studies like the above, the reported data with respect to *BRCA2* PGV frequency does suggest a possible link of germline heterozygous *BRCA2* PGVs with particular cancers occurring in the paediatric and young adult population. However, to further prove causality additional studies would be required to replicate these findings and investigate the mechanism of cancer causation in these non-HBOC tumours. *BRCA1* PGVs were identified in the germline of children diagnosed with what appeared to be non-syndromic neuroblastoma, and also in treatment related secondary malignancies [59]. However, the detected frequency of *BRCA1* variants is too small to support a causative relation. Importantly, the detected variants in the affected children were not always associated with a significant family history indicative of HBOC.

## 5. *BRCA1/2* PGVs in FA-Associated, and Non-Syndromic Cancer: Implications for Management

The management of families affected with FA caused by *BRCA1/2* mutations needs a multidisciplinary approach and include genetic counselling. As affected families in particular with *BRCA2* PGVs do not always have a family history typical of HBOC, an individual approach needs to be taken with respect to genetic carrier-testing and counselling of the wider families of affected individuals [67,68]. Screening for FA-associated organ involvement in affected individuals needs to include central nervous, renal, heart and endocrine assessment as for other types of FA and was reviewed previously elsewhere [1,69,70]. Cancer surveillance in children affected by bi-allelic *BRCA1/2* variants needs to recognise the association with embryonal tumours of childhood and include imaging for brain and other solid malignancies. From the clinical data available, for the *BRCA2*-, but probably not for the *BRCA1*-associated FA-like syndrome, frequent haematological surveillance is important. Given the underlying chromosomal instability, radiation should be avoided where possible with imaging studies preferably undertaken by MRI or USS as appropriate. Treatment of cancer in *BRCA2*-associated FA has been disappointing and the reported survival is poor [21,22,24,27,31,32]. However, it is of note that in both *BRCA1* and *BRCA2*-associated FA-like cases cytotoxic treatment does not always clinically result in excessive toxicity [50,71], and also that malignancies associated with biallelic *BRCA2* PGVs do not always appear particularly chemo-sensitive [71]. For HBOC-associated adult tumours, in which commonly BRCA function is completely lost in tumour development, poly-ADP-ribose-polymerase (PARP)-inhibition, which exploits the acquired DNA repair defect in associated epithelial cancers, has been shown to confer improved survival [72]. While in individuals with bi-allelic germline defects in the *BRCA* genes the tumour-selectivity of these drugs is lost as every cell will be PARP-inhibition sensitive, characteristics of “*BRCAness*” have been identified as potential therapeutic vulnerabilities, and PARP-inhibition is explored as a therapeutic modality in apparently sporadic childhood and young persons’ cancers [73,74]. To what extent this feature is related to germline variants is not fully determined.

## 6. Summary and Perspectives

The discovery of *BRCA1/2* pathogenic variants underlying FA-like disorders puts the efforts of understanding FA on a cellular level in the context of the DNA damage response, and clinically in the context of inherited cancer predisposition. The spectrum and patterns of developmental abnormalities in BRCA-associated FA and FA-like disease provide unique insight into the role of BRCA2 and BRCA1 proteins and their essential but distinct role in development and cancer prevention. The tumour spectrum in these conditions point to the fundamental role of the BRCA genes for development of multiple organ systems, with the intriguing difference with respect to haematopoiesis, which is clinically affected in patients carrying bi-allelic PGVs in *BRCA2*, but not bi-allelic PGVs *BRCA1.* The residual function of BRCA proteins in cases with bi-allelic PGVs is in many cases conferred by protein products of alternatively splicing. Further studies with respect to governance of expression of BRCA-splice variants and functional interaction dynamics of these “spliced” BRCA proteins, which have been detected also in non-FA tissue [37,75,76], will provide further mechanistic insight in the developmental roles and cancer prevention mediated by the BRCA proteins. This could also inform better understanding of acquired chemo-resistance in classical HBOC-associated BRCA-associated cancers. With respect to *BRCA*-variants in non-syndromic childhood malignancies it will be important to further characterise mutations associated with specific cancers, and determine to what extent associated cancers lose the intact allele or if alternative pathways could be involved, as suggested for some tissue-specific *BRCA2*-associated cancers [45,77]. This could be a therapeutic vulnerability for a small and diverse group of sporadic childhood cancers. The complexity of the management with BRCA2 PGVs with non-FA childhood malignancy with respect genetic implications for the wider family of affected individuals with germ line *BRCA2* variants has been illustrated [78], and will need wider considerations. With respect to cancers in FA irrespective of genetic subtype, there is an urgent clinical need for effective cancer treatments that are tolerated with the inherited chromosomal instability, yet effective targeting associated malignant disease. 

## Figures and Tables

**Figure 1 genes-12-01520-f001:**
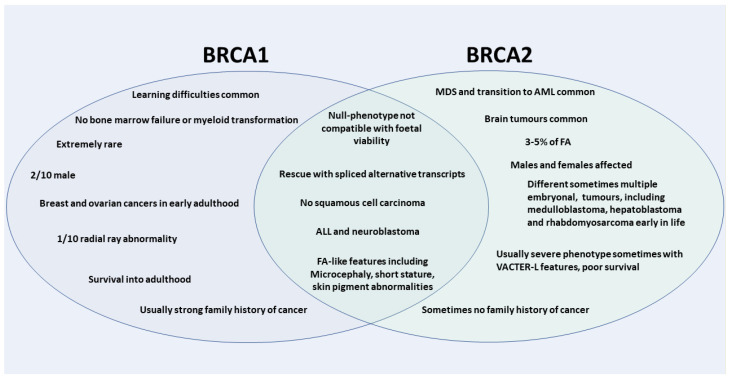
Common and diverse clinical and cellular aspects of bi-allelic pathogenic variants in *BRCA1* and *BRCA2*.

**Table 1 genes-12-01520-t001:** Frequency of *BRCA1* and *BRCA2* variants in large-scale germ line studies in childhood and adolescent malignancies.

	BRCA1 Variants	BRCA2 Variants	Ref
Long Term Survivor Study	14/4402	20/4402	Qin et al. 2020 [59]
Osteosarcoma	3/1440	8/1440	Mirabello et al. 2020 [60]
Paediatric Glioma	1/220	1/220	Muskens et al. 2020 [61]
Rhabdomyosarcoma	1/615	6/615	Li et al. 2021 [65]
Paed and adol. non-Hodgkins Lymphoma	not included	13/1380	Wang et al. 2019 [67]
Medulloblastoma	-	11/1022	Waszak et al. 2018 [64]

## Data Availability

Not applicable.

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
