# Peer review of "Fanconi Anaemia, Childhood Cancer and the BRCA Genes"

_genes, 2021, doi:10.3390/genes12101520_

Round 1
Reviewer 1 Report
In their current review manuscript, Emma R. Woodward and Stefan Meyer draw interesting associations between BRCA1/BRCA2 genes and Fanconi anemia and childhood cancer. Although the paper is relatively short, it is well written, and it stimulates the reader to think on this topic further (e.g., BRCA1 and BRCA2 differences and commonalities, implications for mechanisms and cancer biology, diagnosis and management of BRCA-associated FA).
Specific comments:
- In the current version, there are no figures or tables associated with this manuscript. The paper would profit from such graphical representations (for example BRCA1/2 drawings, table(s) listing genetic variants and symptoms) and it would be really useful for the readership, if the authors could think of adding 1 or 2 such elements.
- There are some minor grammar issues and typos (for example ‘homology directed’ should probably be ‘homology-directed’, ‘BRCA-associated’ instead of ‘BRCA associated’, line 161 should be ‘BRCA1 deficiency’ and not ‘BRCA1 deficiently’ etc.). Also, some repetitions of words/phrases within a single sentence could be avoided (lines 24 and 25 ‘associated’, line 122 twice ‘most cases’).
- Line 121 – the authors write ‘this small but important subgroup’ – would it be possible to somehow numerically indicate how small this subgroup is (either in relation to the entire FA group or in comparison to BRCA2-caused FA syndrome)?
Author Response
In their current review manuscript, Emma R. Woodward and Stefan Meyer draw interesting associations between BRCA1/BRCA2 genes and Fanconi anemia and childhood cancer. Although the paper is relatively short, it is well written, and it stimulates the reader to think on this topic further (e.g., BRCA1 and BRCA2 differences and commonalities, implications for mechanisms and cancer biology, diagnosis and management of BRCA-associated FA).
We thank this reviewer for his positive assessment and we are pleased that we appear to have been able to deliver our messages in a palatable format.
Specific comments:
- In the current version, there are no figures or tables associated with this manuscript. The paper would profit from such graphical representations (for example BRCA1/2 drawings, table(s) listing genetic variants and symptoms) and it would be really useful for the readership, if the authors could think of adding 1 or 2 such elements.
We agree with this suggestion, and we have provided two elements that illustrate the narrative. We agree with this reviewer that this makes the piece easier to access. Please see the revised file for details.
There are some minor grammar issues and typos (for example ‘homology directed’ should probably be ‘homology-directed’, ‘BRCA-associated’ instead of ‘BRCA associated’, line 161 should be ‘BRCA1 deficiency’ and not ‘BRCA1 deficiently’ etc.). Also, some repetitions of words/phrases within a single sentence could be avoided (lines 24 and 25 ‘associated’, line 122 twice ‘most cases’).
This is addressed in the revised version. We have changed the wording to avoid repetition.
Line 121 – the authors write ‘this small but important subgroup’ – would it be possible to somehow numerically indicate how small this subgroup is (either in relation to the entire FA group or in comparison to BRCA2-caused FA syndrome)?
We agree it is important to attach a number – we have done that.
Reviewer 2
This review by Emma et.al summarized BRCA1/2 pathogenic mutations associated FA (Fanconi anaemia) and non-FA childhood cancer.
But there are some issues which I would like to see addressed:
- The causes of FA are related to at least 15 genes, and 80-90% of FA are caused by mutations in one of these three genes FANCA, FANCC and FANCG. Therefore, what’s the significance of your review about FA and BRCA genes?
The special issue is about the BRCA genes, and we provide a review about aspects of the BRCA genes in FA and sporadic childhood cancer. We include recent novel clinical observations with respect to BRCA2 and BRCA1-associated FA. We have changed the phrasing in the abstract to emphasise this
- The BRCA1 and BRCA2 pathogenic variants leading to FA and childhood cancer has been reviewed before, what’s the difference with other reviews?
Here we include more recent publications (e.g. Radulovic, It al (2021)), and the role of germ line variants in BRCA genes for sporadic cancers, which are all studies from this year or rennet years. We have changed the phrasing in the abstract accordingly
The clinical implication of this review should be highlighted here.
We agree that clinical aspects are important, and we would argue that we address these sufficiently in the section “BRCA1/2 PGVs in FA-associated, and non-syndromic cancer: Implications for management”.
Reviewer 2 Report
This review by Emma et.al summarized BRCA1/2 pathogenic mutations associated FA (Fanconi anaemia) and non-FA childhood cancer.
But there are some issues which I would like to see addressed:
- The causes of FA are related to at least 15 genes, and 80-90% of FA are caused by mutations in one of these three genes FANCA, FANCC and FANCG. Therefore, what’s the significance of your review about FA and BRCA genes?
- The BRCA1 and BRCA2 pathogenic variants leading to FA and childhood cancer has been reviewed before, what’s the difference with other reviews?
- The clinical implication of this review should be highlighted here.
Author Response
This review by Emma et.al summarized BRCA1/2 pathogenic mutations associated FA (Fanconi anaemia) and non-FA childhood cancer.
But there are some issues which I would like to see addressed:
- The causes of FA are related to at least 15 genes, and 80-90% of FA are caused by mutations in one of these three genes FANCA, FANCC and FANCG. Therefore, what’s the significance of your review about FA and BRCA genes?
The special issue is about the BRCA genes, and we provide a review about aspects of the BRCA genes in FA and sporadic childhood cancer. We include recent novel clinical observations with respect to BRCA2 and BRCA1-associated FA. We have changed the phrasing in the abstract to emphasise this.
- The BRCA1 and BRCA2 pathogenic variants leading to FA and childhood cancer has been reviewed before, what’s the difference with other reviews?
Here we include more recent publications (e.g. Radulovic, It al (2021)), and the role of germ line variants in BRCA genes for sporadic cancers, which are all studies from this year or recent years. We have changed the phrasing in the abstract accordingly
The clinical implication of this review should be highlighted here.
We agree that clinical aspects are important, and we would argue that we address these sufficiently in the section “BRCA1/2 PGVs in FA-associated, and non-syndromic cancer: Implications for management”.